# An Explanatory Model of Sport Motivation, Physical Self-Concept and Anxiety as a Function of the Degree of Adherence to the Mediterranean Diet in Future Physical Education Teachers

**DOI:** 10.3390/ijerph192013547

**Published:** 2022-10-19

**Authors:** Eduardo Melguizo-Ibáñez, Pilar Puertas-Molero, Gabriel González-Valero, José Manuel Alonso-Vargas

**Affiliations:** Department of Didactics of Musical, Plastic and Corporal Expression, University of Granada (Spain), Campus de Cartuja, 18071 Granada, Spain

**Keywords:** Mediterranean diet, anxiety, sport motivation, physical education teachers

## Abstract

The present research has the objectives of establishing the relationship between motivational climate towards sport, anxiety, and physical self-concept, and identifying the existing relationships between anxiety, motivational climate, and physical self-concept, broken down into (a) developing an explanatory model of the motivational climate towards sport and its relationship with anxiety and physical self-concept, and (b) contrasting the structural model by means of a multi-group analysis according to the degree of adherence to the Mediterranean diet. A quantitative, comparative, non-experimental (ex post facto), cross-sectional study was carried out with a total sample of 556 participants (23.06 ± 6.23). The instruments used were an ad hoc questionnaire and the Spanish versions of the Perceived Motivational Climate in Sport Questionnaire 2, Beck Anxiety Inventory, Self-Concept Form-5, and the PREDIMED questionnaire. The data reveal that participants who show high adherence to the Mediterranean diet attain higher scores on physical self-concept and anxiety, as well as ego-climate. As a conclusion, it is observed that participants who show a high adherence to the Mediterranean diet show higher scores in physical self-concept and anxiety, as well as in all the variables that make up the ego-climate.

## 1. Introduction

The multiple benefits of regular physical activity have increased interest in physical activity [1]. In this line, the benefits of an active lifestyle are psychological, social, and physiological, such as the reduction of cardiovascular diseases, improvements in mental well-being, and improvements in socio-affective relationships [2]. Despite the various benefits, during adolescence, teenagers are less interested in physical activities due to a preference for more sedentary activities [3]. In view of these findings, ref. [4] states that the role of the physical education teacher is key in the creation of an active lifestyle, due to the motivation they provide for the development of physical education classes.

One of the factors studied in sport psychology is motivation [5]. This concept can be defined as a mechanism that controls the direction and intensity of efforts due to its great potential to explain different human behaviours [6,7]. Among the most studied theories in the motivational field is the achievement goal theory [8]. This theory includes the concept of motivational climate, which can be defined as a set of indicators that people perceive in their environment, and through which they define the failure or success of a given task [9]. Extrapolating this theory to the physical-sports environment, two motivational climates can be found. In the first one, values, such as fun or personal satisfaction (task climate), gain importance, while in the second, extrinsic values are highlighted, fostering competition (ego climate) [10], generating increased levels of frustration and anxiety when the established objectives are not achieved [11,12].

Another factor studied in sports psychology is anxiety. This term is defined as a negative psycho-emotional state characterised by the manifestation of negative states, such as nervousness and worry, directly affecting the somatic and cognitive areas [13,14]. Likewise, motivational climate can help to reduce this disruptive state. In this case, when motivation is oriented towards the task climate, anxiety levels decrease, due to the segregation of neurotransmitters [15,16], however, when sport practice is oriented towards the ego climate, anxiety levels increase, as participants focus on the competition, where success is defined as overcoming rivals and demonstrating superior ability [17]. In this case, continued exposure to such high levels of anxiety has a negative impact on people’s health, because high levels of anxiety have been shown to negatively affect adherence to a healthy dietary pattern, such as the Mediterranean diet [18].

The Mediterranean diet is currently conceived as a healthy dietary pattern not only because of the intake of the foods that make up this pattern, but also because of the quality and cooking of the food and the various health benefits that it provides [19]. The foods that characterise this dietary pattern are those originating from the Mediterranean area, such as olive oil, cereals, fruit, vegetables, and pulses, accompanied by a balanced consumption of eggs, dairy products, and fish [20]. Positive adherence has health benefits, such as reduced cardiovascular disease, prevention of various cancers, improved emotional, and cognitive functioning, and improvements in physical self-concept [21].

Physical self-concept is defined as the perception that subjects have of their own physical appearance [22]. Regular physical exercise has a positive impact on people’s mental self-image [23]. At the same time, body dissatisfaction is one of the factors that affects the development of anxiety levels [24]. Subjects who do not accept themselves physically as they are present higher levels of anxiety, due to the pressure exerted by the media for not complying with the standards of beauty established by different societies [25].

Therefore, the following research hypotheses are presented:

**Hypothese 1 (H1).** 
*Participants who show optimal adherence to the Mediterranean diet are expected to have a better relationship between physical self-concept, motivational climate, physical self-concept, and anxiety than young people who show low or medium adherence.*


**Hypothese 2 (H2).** 
*Participants who show low or medium adherence to the Mediterranean diet are expected to have a worse relationship between physical self-concept, motivational climate, physical self-concept, and anxiety than young people showing optimal adherence.*


Finally, taking into account all that has been developed above, the present research has the general objectives of establishing the relationship between motivational climate towards sport, anxiety, and physical self-concept, and identifying the existing relationships between anxiety, motivational climate, and physical self-concept, broken down into (a) developing an explanatory model of the motivational climate towards sport and its relationship with anxiety and physical self-concept, and (b) contrasting the structural model by means of a multi-group analysis according to the degree of adherence to the Mediterranean diet.

## 2. Materials and Methods

### 2.1. Design and Participants

A descriptive non-experimental (ex post facto) cross-sectional study was carried out on students of the Faculty of Education Sciences of the University of Granada. The study sample consisted of 556 students, 75% female (n = 417) and 25% male (n = 139). The age of the participants was between 18 and 24 years (21.06 ± 6.23). In this case, the subjects participated voluntarily after being informed of the objectives and nature of the study, giving their written and informed consent. In terms of sampling error, a sampling error of 0.05 was assumed, taking into account random sampling by natural groups, giving an error of 0.048, with a confidence interval of 95%.

### 2.2. Instruments and Variables

**Socio-demographic questionnaire:** This is a self-prepared sheet designed to collect socio-demographic variables, such as gender (male or female) and age.

**Mediterranean Diet Prevention Questionnaire [26]:** The present study used the Spanish version [27]. This instrument is composed of 14 items, where once answered, a final score is obtained that categorises participants’ responses into three levels: low adherence (≤7), medium adherence (8–10), and high adherence (≤10). For this research, Cronbach’s Alpha obtained a score of α = 0.815.

**Perceived Motivational Climate in Sport Questionnaire (PMCSQ-2) [28]:** The Spanish version was used in the present research [29]. This instrument consists of 33 items rated on a five-level Likert scale (1 = strongly disagree and 5 = strongly agree), and assesses motivation within two dimensions: task climate (consisting of three sub-dimensions: effort, improvement, and cooperative learning), and ego climate (consisting of three sub-levels: unequal recognition, punishment for mistakes, and rivalry between members). The internal reliability of the task climate was 0.948, while that of the ego climate was 0.966.

**Beck Anxiety Inventory [30]:** The Spanish version was used in the present research [31]. This questionnaire is composed of a total of 21 items, which are measured on a four-level Likert-type scale (0 = not at all and 3 = very much). For this research, Cronbach’s Alpha obtained a score of 0.956.

**Self-Concept Questionnaire Form 5 [32]:** It consists of a total of 30 items, assessed on a Likert scale with values ranging from 1 (Never) to 5 (Always). For the present research, only the items that make up the physical self-concept (5,10,15,20,25,30) were used. The reliability obtained for this questionnaire was α = 0.885.

### 2.3. Procedure

The first step was to carry out a bibliographic search to find out more about the problem in question. Afterwards, from the Department of Didactics of Musical, Plastic and Corporal Expression of the University, a Google Form was created with the instruments described above and the informed consent of the participants. Due to the health situation resulting from the COVID-19 pandemic, the virtual medium was used to send out the questionnaires. In addition, two questionnaires were duplicated to ensure that they were not filled out randomly, but 15 questionnaires were deleted as they were incorrectly filled out. In terms of ethics, the principles set out in the 1975 Declaration of Helsinki were followed at all times, guaranteeing anonymity, as well as the rights of the participants. Finally, the ethics committee 2966/CEIH/2022 of the University of Granada approved the present research.

### 2.4. Data Analysis

IBM SPSS Statics 25.0 software (IBM Corp., Armonk, NY, USA) was used for the statistical analysis of the results. A descriptive analysis of the data was carried out using frequencies and means. Subsequently, a comparative analysis was carried out, using a one-factor ANOVA, where statistically significant differences were determined by means of Pearson’s Chi-Square test, establishing the reliability index at 95%. Subsequently, a comparative analysis was carried out, using a one-factor ANOVA, where statistically significant differences were determined by means of Pearson’s Chi-Square test, establishing the reliability index at 95%. The magnitude of the effect size difference (ES) was obtained with Cohen’s standardised d-index [33] interpreted as null (0.0–0.19), small (0.20–0.49), medium (0.50–0.79), and large (≥0.80). Finally, for the study of the normality of the data, the Kolmogorov-Smirnov test was used, obtaining a normal distribution.

For the structural equation models, the IBM SPSS Amos 26.0 software (IBM Corp., Armonk, NY, USA) was used to establish the relationships between the variables that make up the theoretical model (Figure 1). In this case, a model has been developed for each of the degrees of adherence to the Mediterranean diet. Each of the models is composed of two exogenous variables (TC; EC) and eight endogenous variables (IR; EI; CL; PM; UR; MR; ANX; P-SC). For the endogenous variables, a causal explanation has been made on the basis of the observed associations between the indicators and the degree of measurement reliability. At the same time, the one-way arrows represent the lines of influence between the latent variables and are interpreted from the regression weights. A significance level of 0.05 was established using Pearson’s Chi-Square test.

Observing the model developed, it can be seen that ego-climate and task-climate have an impact on anxiety and physical self-concept.

Finally, the goodness of fit should be evaluated on the Chi-Square, whose values associated with p and non-significant indicate a good fit of the model. In this case, the comparative fit index (CFI; values above 0.95 indicate a good model fit), the goodness-of-fit index (GFI; values above 0.90 indicate an acceptable fit), the incremental reliability index (IFI; values above 0.90 indicate an acceptable fit), and the root mean square approximation (RMSEA; values below 0.1 indicate an acceptable model fit) [34,35,36,37].

## 3. Results

Table 1 shows the results obtained from the comparative analysis. In terms of physical self-concept, participants who show high adherence (M = 3.25) have higher scores than those who show medium (M = 3.17) or low adherence (M = 3.09). Continuing with anxiety, it is observed that participants with high adherence (M = 1.25) obtain higher scores than those with medium (M = 0.83) or low adherence (M = 0.76). Likewise, for cooperative learning, effort/improvement, and important role, higher scores are obtained by participants with low adherence (M = 4.03; M = 4.00; M = 4.10) compared to those with optimal adherence (M = 3.75; M = 3.46; M = 3.70) or medium adherence (M = 4.00; M = 3.85; M = 3.96). Finally, for punishment for mistakes, unequal recognition, and member rivalry, it is observed that participants with optimal adherence (M = 2.75; M = 2.68; M = 3.41) obtain better scores than those with low (M = 2.39; M = 2.66; M = 2.69) or medium adherence (M = 2.37; M = 2.57; M = 2.76).

Moving on to structural equation modelling, the proposed model developed for participants showing low adherence to the Mediterranean diet showed a good fit for all indices. In this case, the Chi-Square analysis showed a non-significant *p*-value ($X^{2}=$ 56.385; df = 16; pl = 0.000), but these data cannot be interpreted independently due to the influence of susceptibility and sample size [38], so other standardised fit indices have been used. The comparative fit index (CFI) analysis obtained a value of 0.969. The normalised fit index (NFI) analysis obtained a value of 0.958, the incremental fit index (IFI) was 0.969, and the Tucker-Lewis index (TLI) obtained a value of 0.946. In addition, the root mean square error of approximation analysis (RMSEA) also obtained a value of 0.076.

Table 2 and Figure 2 show the results obtained for the model developed for participants showing low adherence to the Mediterranean diet. In this case, a negative relationship is observed between task climate and anxiety (r = −0.115), and a positive relationship between ego climate and anxiety (*p* ≤ 0.05; r = 0.199). Likewise, in terms of task climate, positive relationships were observed with important role (*p* ≤ 0.001; r = 0.882), effort/improvement (*p* ≤ 0.001; r = 0.874), cooperative learning (*p* ≤ 0.001; r = 0.853), and physical self-concept (*p* ≤ 0.001; r = 0.345). Regarding ego climate, positive relationships are observed with punishment for mistakes (r = 0.848), unequal recognition (*p* ≤ 0.001; r = 0.916), member rivalry (*p* ≤ 0.001; r = 0.540), and physical self-concept (*p* ≤ 0.05; r = 0.211), however, negative relationships are found with task climate (*p* ≤ 0.05; r = −0.569). Finally, negative relationships are shown between anxiety and physical self-concept (*p* ≤ 0.001; r= −0.353).

The model developed for medium adherence showed a good fit for all indices. In this case, the Chi-Square analysis showed a non-significant *p*-value (X^2^ 40.871; df = 16; pl = 0.001). The comparative fit index (CFI) analysis obtained a value of 0.966. The normalised fit index (NFI) analysis obtained a value of 0.946, the incremental fit index (IFI) was 0.966 and the Tucker-Lewis index (TLI) obtained a value of 0.940. In addition, the root mean square error of approximation analysis (RMSEA) also obtained a value of 0.079.

Table 3 and Figure 3 show the scores obtained for participants showing average adherence to the Mediterranean diet. Regarding anxiety, a negative relationship is observed with task climate (r = −0.117), however, a positive relationship is obtained with ego climate (r = 0.127). For task climate, positive relationships are shown with important role (r = 0.939), effort/improvement (*p* ≤ 0.001; r = 0.806), cooperative learning (*p* ≤ 0.001; r = 0.885), and social self-concept (r = 0.109). In terms of ego climate, positive relationships are obtained with punishment for mistakes (r = 0.822), unequal recognition (*p* ≤ 0.001; r = 0.854), member rivalry (*p* ≤ 0.05; r = 0.635), and physical self-concept (r = 0.013), however, a negative relationship is obtained with task climate (*p* ≤ 0.001; r = 0.870). Lastly, for physical self-concept, a negative relationship is obtained with anxiety (*p* ≤ 0.001; r = −0.265).

The model developed for participants showing optimal adherence to the Mediterranean diet showed a good fit for each of the indices. In this case, the Chi-Square analysis showed a non-significant *p*-value ($X^{2}=$ 12.044; df = 16; pl = 0.741). The comparative fit index (CFI) analysis obtained a value of 0.995. The normalised fit index (NFI) analysis obtained a value of 0.919, the incremental fit index (IFI) was 0.990 and the Tucker-Lewis index (TLI) obtained a value of 0.950. In addition, the root mean square error of approximation analysis (RMSEA) also obtained a value of 0.020.

Table 4 and Figure 4 show the scores obtained for participants showing high adherence to the Mediterranean diet. For anxiety, a positive relationship is observed with task climate (r = 0.031), and ego climate (r = 0.379). For task climate, positive relationships are shown with important role (r = 0.961), effort/improvement (*p* ≤ 0.001; r = 0.961), and cooperative learning (*p* ≤ 0.001; r = 0.867), however, a negative relationship is shown with physical self-concept (*p* ≤ 0.05; r = −0.465). Regarding ego climate, positive relationships are obtained with punishment for mistakes (r = 0.962), unequal recognition (*p* ≤ 0.001; r = 0.851), and member rivalry (*p* ≤ 0.05; r = −0.521), however, a negative relationship is shown with physical self-concept (*p* ≤ 0.05; r = −0.610) and with task climate (*p* ≤ 0.05; r = −0.655). Finally, a negative relationship was observed between physical self-concept and anxiety (r = −0.120).

## 4. Discussion

The present research shows the relationships between the motivational climate developed towards physical activity, anxiety, and physical self-concept as a function of the degree of adherence to a healthy dietary pattern. In this way, the results obtained respond to the proposed objectives, so that the present discussion follows the line of comparing the results obtained with those already obtained in other studies.

In terms of physical self-concept, participants with a high adherence to the Mediterranean diet showed higher scores. These results were similar to those found by Zurita-Ortega et al. [39], Pérez-Marmol et al. [40], and Padial-Ruz et al. [41], stating that young people who engage in regular physical activity tend to consume a nutritious and healthy diet, which has a positive impact on sports performance, as well as on body image and well-being.

Continuing with anxiety, it is observed that participants who reflect a high degree of adherence to the Mediterranean diet show higher scores than those who report a medium or low degree of adherence. Very distant results were obtained by Trigueros et al. [42] affirming Martínez-Rodríguez et al. [43] that healthy eating has a positive impact on the channelling of disruptive states, such as anxiety. Despite these findings, Melguizo-Ibáñez et al. [44] point to a negative process of emotional eating, in which excessive intake of high-calorie foods is used as a coping method in the face of high anxiety. Furthermore, Marchena et al. [18] state that this process can become a behaviour-reinforcing element, damaging health.

Looking at the variables that make up the task climate, it is observed that participants who show low adherence to the Mediterranean diet show higher scores than those who claim to have a medium or high adherence. Similar results were obtained by González-Valero et al. [45] and Balaguer et al. [46], who stated that the main reason for these results is that people who orientate their physical activity towards factors are not concerned about the care of their diet, as they do not require a high sporting performance. On the other hand, regarding the variables that make up the ego-climate, it is observed that those participants who show a high degree of adherence to the Mediterranean diet show higher scores. Similar results were obtained by González-Valero et al. [47], affirming Chacón-Cuberos et al. [48], that when the practice is oriented towards sports performance, greater care of various healthy factors is required.

Based on the results obtained from the equation models, it is observed that for participants with medium and low adherence, there is a negative relationship between anxiety and task climate, but for participants with optimal adherence, there is a positive relationship between the two variables. In view of these findings, Baldó-Vela and Bonfanti [49] state that weight loss disorders lead to food restriction in order to lose weight quickly, generating an increase in anxiety levels [50].

Likewise, looking at the relationships between physical self-concept and motivational climate, positive relationships are observed for low and medium adherence, but negative relationships are obtained for participants who show optimal adherence. Very distant results were found by González-Valero et al. [45] where Conde-Pipó et al. [51] state that a positive motivation towards activity practice has a positive impact on people’s physical self-image. However, Puertas-Molero et al. [24] and Marfil-Carmona et al. [25] point to the media as a discontent with the mental image that the person has of him/herself, affecting physical and mental health [52].

Finally, for the relationship between anxiety and physical self-concept, a negative relationship was observed, regardless of the degree of adherence to the Mediterranean diet. Similar results were obtained by Tóth et al. [53], who stated that disruptive states negatively affect the development of self-concept, as, according to Mascret et al. [23] and Fineschi et al. [54], an unrealistic mental image is projected, which impairs the development of this area of self-concept.

## 5. Limitations and Future Perspectives

The present research reflects a series of limitations. The nature of the study itself is one of them, since being a descriptive and cross-sectional study, it is not possible to establish generalisations in a wider area of the national geography. At the same time, the participants belong to a very specific branch of study. In addition, it should be noted that the data collection was carried out during a period in which a high number of COVID-19 virus infections occurred, which had a negative impact on the total number of participants.

Focusing attention on future perspectives, depending on the results obtained, the aim is to develop an intervention programme where different activities can be used to influence their motivation and to study the effects of this programme on the physical and psychosocial aspects of sport.

## 6. Conclusions

Based on the data from the descriptive analysis, it is observed that participants who show a high adherence to the Mediterranean diet achieve higher scores in physical self-concept, and anxiety, as well as in all the variables that make up the ego-climate. On the other hand, it is observed that participants who show a low adherence to this dietary pattern obtain higher scores in all the variables that make up the task climate.

Following the results obtained in the proposed structural equation models, acceptable values have been obtained for each of the different parameters of the general equation. Regarding the relationship between anxiety and task climate, for participants showing low and medium adherence to the Mediterranean diet, a negative relationship is obtained, while for participants showing high adherence, a positive relationship is obtained. Also, for the relationship between anxiety and ego-climate, a positive relationship was observed for all three levels of adherence to the Mediterranean diet. For the relationship between ego-climate, physical self-concept, task-climate, and physical self-concept, positive relationships are observed for participants who show a low or medium degree of adherence. However, positive relationships are shown for these variables when a high degree of adherence to the Mediterranean diet is claimed. Finally, for the relationship between physical self-concept and anxiety, negative relationships were obtained for the three levels of adherence to the Mediterranean diet.

## Figures and Tables

**Figure 1 ijerph-19-13547-f001:**
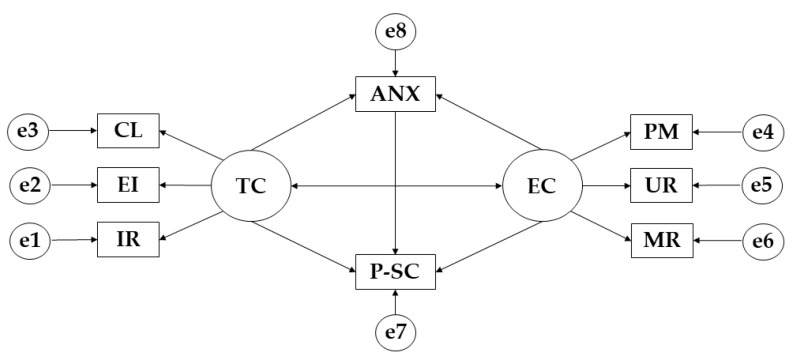
Theoretical Model Proposed. **Note:** Task Climate (TC); Cooperative Learning (CL); Effort/Improvement (EI); Important Role (IR); Ego Climate (EC); Punishment for Mistakes (PM); Unequal Recognition (UR); Rivalry between group members (MR); Physical Self-Concept (P-SC); Anxiety (ANX).

**Figure 2 ijerph-19-13547-f002:**
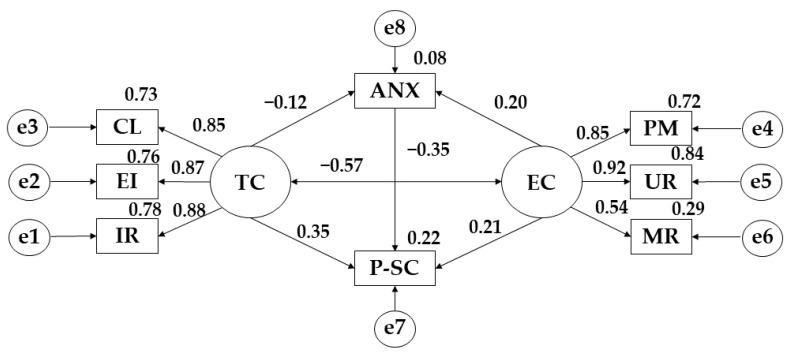
Model developed for low adherence to the Mediterranean Diet. **Note:** Task Climate (TC); Cooperative Learning (CL); Effort/Improvement (EI); Important Role (IR); Ego Climate (EC); Punishment for Mistakes (PM); Unequal Recognition (UR); Member Rivalry (MR); Physical Self-Concept (P-SC); Anxiety (ANX).

**Figure 3 ijerph-19-13547-f003:**
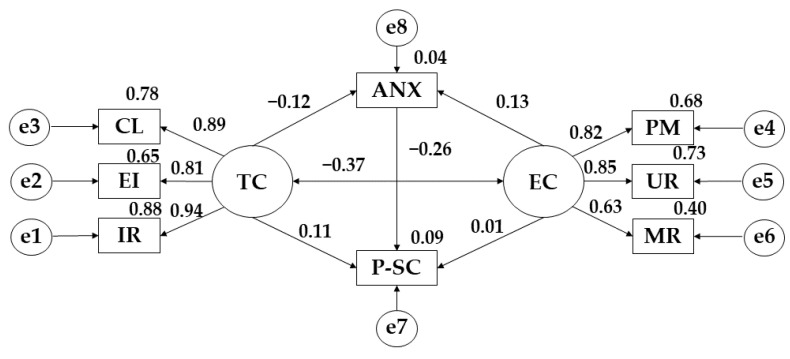
Model developed for medium adherence to the Mediterranean Diet. **Note:** Task Climate (TC); Cooperative Learning (CL); Effort/Improvement (EI); Important Role (IR); Ego Climate (EC); Punishment for Mistakes (PM); Unequal Recognition (UR); Member Rivalry (MR); Physical Self-Concept (P-SC); Anxiety (ANX).

**Figure 4 ijerph-19-13547-f004:**
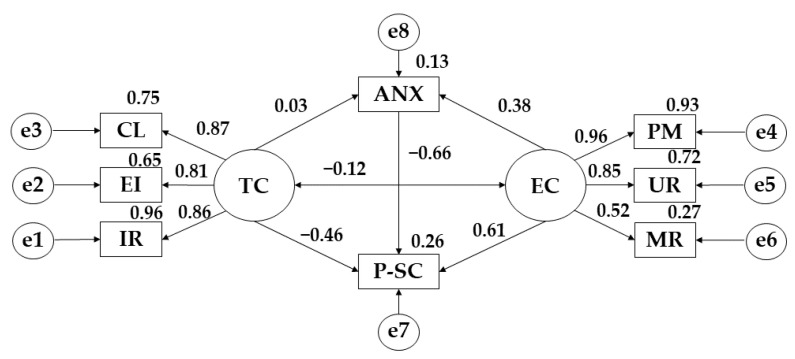
Model developed for high adherence to the Mediterranean diet. **Note:** Task Climate (TC); Cooperative Learning (CL); Effort/Improvement (EI); Important Role (IR); Ego Climate (EC); Punishment for Mistakes (PM); Unequal Recognition (UR); Member Rivalry (MR); Physical Self-Concept (P-SC); Anxiety (ANX).

**Table 1 ijerph-19-13547-t001:** Comparative Study of the Sample.

		N	M	SD	F	*P*	ES(d)	95% CI
**P-SC**	**Low Adherence**	342	3.09	0.82	0.634	≤0.05 ^a,c^	0.113 ^a^ 0.098 ^c^	[0.062; 0.288] ^a^ [0.077; 0.273] ^c^
	**Medium Adherence**	197	3.17	0.81				
	**High Adherence**	17	3.25	0.64				
**ANX**	**Low Adherence**	342	0.76	0.62	1.746	≤0.05 ^b^	0.787 ^b^	[0.297; 1.278]
	**Medium Adherence**	199	0.83	0.65				
	**High Adherence**	17	1.25	0.67				
**CL**	**Low Adherence**	342	4.03	0.86	0.299	˃0.05	NP	NP
	**Medium Adherence**	199	4.00	0.87				
	**High Adherence**	17	3.75	0.61				
**EI**	**Low Adherence**	342	4.00	0.67	3.779	≤0.05 ^c^	0.213 ^c^	[0.038; 0.388]
	**Medium Adherence**	199	3.85	0.76				
	**High Adherence**	17	3.46	0.48				
**IR**	**Low Adherence**	342	4.10	0.80	2.159	˃0.05	NP	NP
	**Medium Adherence**	199	3.96	0.89				
	**High Adherence**	17	3.70	0.66				
**PM**	**Low Adherence**	342	2.39	0.81	0.438	˃0.05	NP	NP
	**Medium Adherence**	199	2.37	0.77				
	**High Adherence**	17	2.75	0.90				
**UR**	**Low Adherence**	342	2.66	1.01	0.371	˃0.05	NP	NP
	**Medium Adherence**	199	2.57	1.04				
	**High Adherence**	17	2.68	1.10				
**MR**	**Low Adherence**	342	2.69	0.90	1.557	˃0.05	NP	NP
	**Medium Adherence**	199	2.76	0.90				
	**High Adherence**	17	3.41	1.13				

**Note:**^a^ Differences between High Adherence and Medium Adherence. ^b^ Differences between High Adherence and Low Adherence. ^c^ Differences between Medium Adherence and Low Adherence. **Note:** Task Climate (TC); Cooperative Learning (CL); Effort/Improvement (EI); Important Role (IR); Ego Climate (EC); Punishment for Mistakes (PM); Unequal Recognition (UR); Member Rivalry (MR); Physical Self-Concept (P-SC); Anxiety (ANX).

**Table 2 ijerph-19-13547-t002:** Structural model for low adherence to the Mediterranean Diet.

Associations between Variables	R.W.				S.R.W.
	Estimations	S.E.	C.R.	*p*	Estimations
ANX ← TC	−0.101	0.061	−1.645	0.100	−0.115
ANX ← EC	0.180	0.064	2.810	**	0.199
IR ← TC	1.000				0.882
EI ← TC	0.834	0.040	20.933	***	0.874
CL ← TC	1.042	0.052	20.214	***	0.853
PM ← EC	1.000				0.848
UR ← EC	1.354	0.081	16.798	***	0.916
MR ← EC	0.713	0.070	10.242	***	0.540
P-SC ← TC	0.396	0.076	5.240	***	0.345
P-SC ← EC	0.250	0.079	3.163	**	0.211
P-SC ← ANX	−0.461	0.066	−6.983	***	−0.353
EC ←→ TC	−0.278	0.036	−7.796	***	−0.569

**Note 1:** Regression Weights (R.W.); Standardised Regression Weights (S.R.W.); Estimation error (S.E.); Critical Ratio (C.R.) **Note 2:** Task Climate (TC); Cooperative Learning (CL); Effort/Improvement (EI); Important Role (IR); Ego Climate (EC); Punishment for Mistakes (PM); Unequal Recognition (UR); Rivalry between group members (MR); Physical Self-Concept (P-SC); Anxiety (ANX). **Note 3:** ** *p* ≤ 0.05; *** *p* ≤ 0.001.

**Table 3 ijerph-19-13547-t003:** Structural model for medium adherence to the Mediterranean diet.

Associations between Variables	R.W.				S.R.W.
	Estimations	S.E.	C.R.	*p*	Estimations
ANX ← TC	−0.091	0.060	−1.524	0.128	−0.117
ANX ← EC	0.130	0.083	1.576	0.115	0.127
IR ← TC	1.000				0.939
EI ← TC	0.734	0.047	15.741	***	0.806
CL ← TC	0.923	0.050	18.536	***	0.885
PM ← EC	1.000				0.822
UR ← EC	1.392	0.131	10.602	***	0.854
MR ← EC	0.904	0.101	8.942	***	0.635
P-SC ← TC	0.105	0.073	1.445	0.148	0.109
P-SC ← EC	0.016	0.101	0.163	0.870	0.013
P-SC ← ANX	−0.331	0.084	−3.929	***	−0.265
EC ←→ TC	−0.197	0.045	−4.337	***	−0.367

**Note 1:** Regression Weights (R.W.); Standardised Regression Weights (S.R.W.); Estimation error (S.E.); Critical Ratio (C.R.) **Note 2:** Task Climate (TC); Cooperative Learning (CL); Effort/Improvement (EI); Important Role (IR); Ego Climate (EC); Punishment for Mistakes (PM); Unequal Recognition (UR); Rivalry between group members (MR); Physical Self-Concept (P-SC); Anxiety (ANX). **Note 3:** ** *p* ≤ 0.05; *** *p* ≤ 0.001.

**Table 4 ijerph-19-13547-t004:** Structural model for high adherence to the Mediterranean Diet.

Associations between Variables	R.W.				S.R.W.
	Estimations	S.E.	C.R.	*p*	Estimations
ANX ← TC	0.027	0.204	0.131	0.896	0.031
ANX ← EC	0.322	0.208	1.548	0.122	0.379
IR ← TC	1.000				0.961
EI ← TC	0.756	0.118	6.421	***	0.807
CL ← TC	0.921	0.123	7.493	***	0.867
PM ← EC	1.000				0.962
UR ← EC	1.051	0.161	6.544	***	0.851
MR ← EC	0.552	0.175	3.156	**	0.521
P-SC ← TC	−0.452	0.227	−1.991	**	−0.465
P-SC ← EC	−0.595	0.246	−2.416	**	−0.610
P-SC ← ANX	−0.138	0.197	−0.700	0.484	−0.120
EC ←→ TC	−0.492	0.170	−2.902	**	−0.655

**Note 1:** Regression Weights (R.W.); Standardised Regression Weights (S.R.W.); Estimation error (S.E.); Critical Ratio (C.R.) **Note 2:** Task Climate (TC); Cooperative Learning (CL); Effort/Improvement (EI); Important Role (IR); Ego Climate (EC); Punishment for Mistakes (PM); Unequal Recognition (UR); Rivalry between group members (MR); Physical Self-Concept (P-SC); Anxiety (ANX). **Note 3:** ** *p* ≤ 0.05; *** *p* ≤ 0.001.

## Data Availability

The data used to support the findings of current study are available from the corresponding author upon request.

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
