# Peer review of "An Explanatory Model of Sport Motivation, Physical Self-Concept and Anxiety as a Function of the Degree of Adherence to the Mediterranean Diet in Future Physical Education Teachers"

_ijerph, 2022, doi:10.3390/ijerph192013547_

Round 1
Reviewer 1 Report
Its really an interesting study.
However, there are still many dietary patterns with health benefit, it seems that author can give more illustration about why use Meditarranean diet as the main role.
How to make differentiate Mediterranean diet from other healthy diets that elevates sport motivation, physical self-concept, etc...?
that is, people with healthy diet nay lead to healthy lifestyle or helathy mindset, what makes Meditarranean unique in this study?
Did author evaluate he impact of covid-19 in the study?
after all, there're still promissing results and excellent discussion in the amnuscript.
Author Response
Comment 1
Its really an interesting study.
However, there are still many dietary patterns with health benefit, it seems that author can give more illustration about why use Meditarranean diet as the main role.
Response 1
Thank you very much for your suggestion. In this case the study has been developed with participants close to the Mediterranean area, specifically Andalusia (Spain). Within this geographical region, it is very common to find varied dishes, which are made with food through such a dietary pattern. In order for you to better understand the importance of the Mediterranean diet in this Spanish region, a book with typical Andalusian dishes is attached, highlighting the importance of the Mediterranean diet: https://www.juntadeandalucia.es/export/drupaljda/Cocina_Andaluza_Dieta_Mediterranea_1995.pdf
Comment 2
How to make differentiate Mediterranean diet from other healthy diets that elevates sport motivation, physical self-concept, etc...?
Response 2
Thank you very much for your comment. In this case the reason that differentiates the Mediterranean diet is indicated in the text, however it is stated below "The foods that characterise this dietary pattern are those originating from the median zone, such as olive oil, cereals, fruits, vegetables and legumes, accompanied by a balanced consumption of eggs, dairy and fish [20]. Positive adherence has health benefits, such as reduced cardiovascular disease, prevention of several types of cancer, improved emotional and cognitive functioning and improved physical self-concept [21]".
Comment 3
That is, people with healthy diet nay lead to healthy lifestyle or healthy mindset, what makes Mediterranean diet unique in this study?
Response 3
Thank you very much for your comment. The authors certainly see this issue as the core of this study. In this case, the study is based on the study of how adherence to the Mediterranean diet can help to improve psychological aspects such as physical self-concept and motivational climate. Positive adherence has been shown to have physical health benefits (reduction of blood pressure, reduction of waist circumference, among others) but the psychological domain has been little studied. This is why the present research aims to study whether positive adherence to the Mediterranean diet brings benefits in the relationships proposed in the variables that make up the structural equation models.
Comment 4
Did author evaluate the impact of covid-19 in the study?
Response 4
Thank you very much for your comment. In this case the study conducted is supposed to be a pilot study on how COVID-19 confinement has influenced adherence to the Mediterranean diet. Subsequently, a new study will be carried out evaluating the results obtained before confinement and after confinement.
Comment 5
After all, there're still promissing results and excellent discussion in the manuscript.
Response 5
Thank you very much for your comment. The authors have tried to provide a comprehensive discussion that meets the research objectives and hypotheses.

Reviewer 2 Report
The study describes an interesting topic of searching for the relationship between psychoemotional factors related to the perception of physical activity by people who will conduct physical education classes in the future. Another interesting topic is the link between the perception of physical activity and adherence to the principles of the Mediterranean diet as an important element of the lifestyle.
In this context, the obtained results seem interesting, not always indicating directly proportional relationships between the studied characteristics, e.g. people showing greater adherence to the Mediterranean diet were characterized by a higher level of anxiety, and a higher level of anxiety was associated with a lower physical self-concept, regardless of the diet. However, the authors tried to explain the observed relationships.
Detailed comments:
· Wouldn't it be enough to leave hypothesis 3 and 4? Hypothesis 1 and 2 are contained in the next two.
· It is also worth adding an explanation of the abbreviations related to the statistical analysis of the results below the tables.
· Perhaps it is worth adding a table with the results of the questionnaires and the level of implementation of the recommendations of the Mediterranean diet.
· Shouldn't we also add information about the characteristics of the group, e.g. BMI values, level of physical activity?
· In the discussion of the results, there is no reference to the Mediterranean diet as a lifestyle model in which nutritional recommendations are combined with recommendations for behaviour related to physical activity. Which aspects of the diet were more important for people with a high level of adherence to the Mediterranean diet?
Author Response
REVIEWER 2
Comment 1
The study describes an interesting topic of searching for the relationship between psychoemotional factors related to the perception of physical activity by people who will conduct physical education classes in the future. Another interesting topic is the link between the perception of physical activity and adherence to the principles of the Mediterranean diet as an important element of the lifestyle.
In this context, the obtained results seem interesting, not always indicating directly proportional relationships between the studied characteristics, e.g. people showing greater adherence to the Mediterranean diet were characterized by a higher level of anxiety, and a higher level of anxiety was associated with a lower physical self-concept, regardless of the diet. However, the authors tried to explain the observed relationships.
Response 1
Thank you very much for your various comments. The authors believe that the data show novel results with respect to the subject matter.
Comment 2
Wouldn't it be enough to leave hypothesis 3 and 4? Hypothesis 1 and 2 are contained in the next two.
Response 2
Thank you very much for your comment. The authors agree with your comment and we have removed the requested assumptions.
Comment 3
It is also worth adding an explanation of the abbreviations related to the statistical analysis of the results below the tables.
Response 3
Thank you very much for your suggestion. Data related to the type of analysis has been added.
Comment 4
Perhaps it is worth adding a table with the results of the questionnaires and the level of implementation of the recommendations of the Mediterranean diet.
Response 4
Comment 5
Shouldn't we also add information about the characteristics of the group, e.g. BMI values, level of physical activity?
Response 5
Thank you very much for your comment. After discussion, the authors felt that adding another table could confuse the reader as there are a total of 4 tables. Furthermore, it should also be noted that there are 4 other figures, therefore we consider that adding another table may confuse the readers.
Comment 6
In the discussion of the results, there is no reference to the Mediterranean diet as a lifestyle model in which nutritional recommendations are combined with recommendations for behaviour related to physical activity. Which aspects of the diet were more important for people with a high level of adherence to the Mediterranean diet?
Response 6
Thank you very much for your comment. In this case it has been observed that the motivational climate towards which the physical-sports practice originates plays a vital role in the development of an active lifestyle. It has also been observed that adherence to the Mediterranean diet plays a key role in the adherence to a healthy lifestyle.
In this case, according to their suggestion, the most important aspects for a high level of adherence to the Mediterranean diet are a higher consumption of fruit and vegetables, a moderate consumption of red meat, and a consumption of vegetable fats instead of animal fats.
There is also a reduced consumption of industrial pastries, sweets (sweets and sugary drinks) and visits to fast food restaurants.
